# Approximate Value Equivalence

**Christopher Grimm**
Computer Science & Engineering
University of Michigan
crgrimm@umich.edu

**André Barreto**
DeepMind
andrebarreto@deepmind.com

**Satinder Singh**
DeepMind
baveja@deepmind.com

## Abstract

Model-based reinforcement learning agents must make compromises about which aspects of the environment their models should capture. The value equivalence (VE) principle posits that these compromises should be made considering the model's eventual use in value-based planning. Given sets of functions and policies, a model is said to be order-$k$ VE to the environment if $k$ applications of the Bellman operators induced by the policies produce the correct result when applied to the functions. Prior work investigated the classes of models induced by VE when we vary $k$ and the sets of policies and functions. This gives rise to a rich collection of topological relationships and conditions under which VE models are optimal for planning. Despite this effort, relatively little is known about the planning performance of models that fail to satisfy these conditions. This is due to the rigidity of the VE formalism, as classes of VE models are defined with respect to *exact* constraints on their Bellman operators. This limitation gets amplified by the fact that such constraints themselves may depend on functions that can only be approximated in practice. To address these problems we propose approximate value equivalence (AVE), which extends the VE formalism by replacing equalities with error tolerances. This extension allows us to show that AVE models with respect to one set of functions are also AVE with respect to any other set of functions if we tolerate a high enough error. We can then derive bounds on the performance of VE models with respect to *arbitrary sets of functions*. Moreover, AVE models more accurately reflect what can be learned by our agents in practice, allowing us to investigate previously unexplored tensions between model capacity and the choice of VE model class. In contrast to previous works, we show empirically that there are situations where agents with limited capacity should prefer to learn more accurate models with respect to smaller sets of functions over less accurate models with respect to larger sets of functions.

## 1 Introduction

Reinforcement learning (RL) is a general framework in which an agent learns to maximize the reward it receives from its environment by interacting with it [Sutton and Barto, 2018]. Though RL makes no prescriptions about their internal processes, agents are sometimes endowed with the ability to learn simulators of their environment, called *models*, which enable predicting the effects of sequences of actions without executing them in the environment. The subfield of RL which studies such agents is referred to as model-based reinforcement learning (MBRL).

36th Conference on Neural Information Processing Systems (NeurIPS 2022).

In classical MBRL, agents endeavor to learn models that accurately capture the dynamics of their environments and subsequently use these models in their estimation of the values of different policies. Though having an accurate model is sufficient for this task, they are frequently impractical to learn— as MBRL agents are deployed in more complex environments, the demands of learning models which accurately capture these complexities become more severe. Fortunately, such models are not necessary for learning performant MBRL agents, as demonstrated by several recent works [Oh et al., 2017, Silver et al., 2017, Farquhar et al., 2018, Schrittwieser et al., 2020]. The agents in these works learn models that are accurate at what they are eventually used for: the estimation of policy values. In particular, these models are learnt to be able to accurately predict the values of policies in the environment rather than to be able to accurately predict the environment's dynamics.

Grimm et al. [2020, 2021] provide a theoretical underpinning for models learned in this way by establishing the *value equivalence* (VE) principle. VE partitions the space of models an agent can learn into different VE classes according to properties of their Bellman operators. For a given integer $k$, an order-$k$ VE class consists of models whose $k$-step Bellman operators of certain policies match the environment's when applied to certain functions. The policies and functions used as VE constraints are the identifiers of the equivalent classes. Accordingly, much of the prior work on VE has concerned the selection of these sets such that the corresponding VE models are of use to an agent. In particular, Grimm et al. [2021] proved that any model in the infinite-order VE—also known as *proper value equivalent* or PVE—class induced by all deterministic policies is sufficient for optimal planning. However, with the exception of this specific model class and its subsets, the analyses by Grimm et al. [2020, 2021] offer no bounds on the performance of other VE classes.

In this work we extend the VE framework by developing an approximate theory of value equivalence which augments VE classes with an error tolerance parameter that controls the degree to which models can violate their class' constraints (Definition 1). We call the resulting framework *approximate value equivalence* (AVE). While previous work on VE demonstrated topological relationships between specific instances of VE classes (e.g., between classes with order 1 [Grimm et al., 2020] or between classes with fixed $\Pi$ and $\mathcal{V}$ and orders that are multiples of one another [Grimm et al., 2021]), we show that arbitrary pairs of AVE classes can be topologically related by tolerating sufficient error (Proposition 1). This allows us to bound the performance of models in *any* AVE class in terms of the minimum error that must be tolerated in order to topologically relate it to a subset of the aforementioned PVE class (Proposition 2). In Section 4, we proceed by deriving a series of topological relationships between AVE classes and leveraging this connection between topology and performance to produce various performance bounds for AVE classes. We are ultimately able to combine these relationships together to produce a general performance bound over AVE classes with respect to arbitrary orders and function sets— dramatically expanding the set of VE model classes with theoretical guarantees. Finally, we leverage our study of AVE classes to reveal previously overlooked tensions between model capacity and the choice of model class. In particular, we empirically demonstrate situations in which a low-capacity agent achieves higher performance by learning a VE model with respect to *fewer* functions (and tolerating a lower approximation error) than by learning one with respect to more functions (and tolerating a higher approximation error).

## 2   Background

An agent's interaction with its environment is modeled using a Markov Decision Process (MDP) [Puterman, 1994] denoted by $\langle \mathcal{S}, \mathcal{A}, r, p, \gamma \rangle$, where $\mathcal{S}$ is a state-space, $\mathcal{A}$ is a set of actions that can be taken from each state, $r(s, a)$ is the expected reward of taking action $a$ from state $s$, $p(s'|s, a)$ is the probability of transitioning to state $s'$ when action $a$ is taken from state $s$ and $\gamma \in [0, 1]$ is a discount factor. We additionally make the standard assumption that the environment has a finite maximum reward: $r_{\max} \equiv \max_{s,a} r(s, a) < \infty$. The behavior of an agent is specified by a policy, $\pi : \mathcal{S} \to \mathcal{P}(\mathcal{A})$, where $\mathcal{P}(\mathcal{A})$ is the set of probability distributions over $\mathcal{A}$. We denote the set of all policies as $\Pi$ and the set of all functions mapping $\mathcal{S} \mapsto \mathbb{R}$ as $\mathbb{V}$. The performance of a policy in an MDP is measured by its value function:

$$v_\pi(s) \equiv \mathbb{E}_{\pi,p} \left[ \sum_{t=0}^{\infty} \gamma^t r(S_t, A_t) | S_0 = s \right], \tag{1}$$

where $\mathbb{E}_{\pi,p}[\cdot]$ denotes expectation over trajectories generated by policy $\pi$ in an MDP with transition dynamics given by $p$, and $S_t$ and $A_t$ are random variables for the state occupied and action taken by the agent at timestep $t$. We denote the set of all value functions in an environment as $\mathbb{V}_\Pi = \{v_\pi : \pi \in \Pi\}$.

An RL agent's goal is to find a policy with maximal value across all states [Sutton and Barto, 2018]. This process can be carried out by alternating between computing the value of the agent's policy and improving it. Value functions can be computed iteratively using the policy's *Bellman operator*:

$$\mathcal{T}_\pi[v](s) \equiv \mathbb{E}_{A\sim\pi(\cdot|s),S'\sim p(\cdot|s,A)}\left[r(s,A) + \gamma v(S')\right], \tag{2}$$

which accepts a function $v \in \mathbb{V}$ and returns an updated function also in $\mathbb{V}$. We use $\mathcal{T}_\pi^k$ to refer to $k$ applications of $\mathcal{T}_\pi$, also known as the policy's $k$-step Bellman operator. It is known that, in the limit, $k$-step Bellman operators converge to their corresponding value functions: $\lim_{k\to\infty} \mathcal{T}_\pi^k[v] = v_\pi \ \forall v \in \mathbb{V}$. In RL it is assumed that the agent does not have access to $r$ and $p$, and thus $\mathcal{T}_\pi$ cannot be computed directly. MBRL addresses this problem by learning a model, $\tilde{m} \equiv (\tilde{r}, \tilde{p})$, that is used in place of the environment's reward and transition dynamics to construct model Bellman operators, denoted $\tilde{\mathcal{T}}_\pi$. We denote the model's maximum reward as $\tilde{r}_{\max} \equiv \max_{s,a} \tilde{r}(s,a)$.

Traditional MBRL methods attempt to learn models such that $\tilde{r} \approx r$ and $\tilde{p} \approx p$ [Sutton, 1991]. In contrast, there are alternative paradigms in which a model is learned with its eventual use in mind, such as value-aware model learning (VAML; [Farahmand et al., 2017]) and the VE principle [Grimm et al., 2020]. This work concerns the VE principle, under which models are learned to support their eventual use in estimating value functions. Accordingly, VE delineates models according to *VE constraints* defined by policies $\pi \in \Pi$ and functions $v \in \mathcal{V}$. We denote the class of order-$k$ VE models with respect to a set of policies $\Pi \subseteq \mathbb{\Pi}$ and a set of functions $\mathcal{V} \subseteq \mathbb{V}$ as

$$\mathcal{M}^k(\Pi, \mathcal{V}) \equiv \{\tilde{m} \in \mathcal{M} : \tilde{\mathcal{T}}_\pi^k v = \mathcal{T}_\pi^k v \ \forall \pi \in \Pi, \forall v \in \mathcal{V}\}. \tag{3}$$

In the limit of $k \to \infty$, Grimm et al. [2021] showed that order-$k$ VE classes converge to special model classes whose models' value functions $\tilde{v}_\pi = \lim_{k\to\infty} \tilde{\mathcal{T}}_\pi^k v$ match those of the environment:

$$\mathcal{M}^\infty(\Pi) \equiv \{\tilde{m} \in \mathcal{M} : \tilde{v}_\pi = v_\pi \ \forall \pi \in \Pi\}. \tag{4}$$

These limiting model classes are referred to as *proper value equivalent (PVE)* due to their dependence on model and environment *value functions*. The PVE class with respect to all deterministic policies, $\mathbb{\Pi}_{\det}$, is denoted by $\mathcal{M}^\infty(\mathbb{\Pi}_{\det})$ and contains only models which can plan optimally. Notably, this is the only non-trivial model class within the VE framework with such a performance guarantee [Grimm et al., 2021].

We say that a set of functions $\mathcal{V}$ mapping $\mathcal{S} \to \mathbb{R}$ is *bounded* if there exists some $v_{\max} < \infty$ such that $v(s) \leq v_{\max}$ for all $v \in \mathcal{V}$ and $s \in \mathcal{S}$. We also consider a notion of boundedness for model classes, referring to a model class as *bounded* if each model in it has bounded maximum reward: $\tilde{r}_{\max} < \infty$ for each model $\tilde{m}$ in the class.

## 3  Approximate value equivalence

We begin our treatment of the approximate theory of value equivalence by extending the definitions of order-$k$ VE and PVE classes given in Eqs. 3 and 4 to the approximate setting. Recall that the exact versions of these model classes are characterized by sets of policies and functions which induce a set of VE constraints on the models' Bellman operators. In this work we will consider model classes whose models can violate these constraints up to some limited degree, $\epsilon \in \bar{\mathbb{R}}^+$ where $\bar{\mathbb{R}}^+$ denotes the non-negative, extended real numbers: $[0, \infty]$. We say that these *approximate value equivalent (AVE)* classes *tolerate* $\epsilon$ error.

**Definition 1.** *(Approximate Value Equivalence) Given $\Pi \subseteq \mathbb{\Pi}$, $\mathcal{V} \subseteq \mathbb{V}$ and $\epsilon \in \bar{\mathbb{R}}^+$, we denote the set of order-$k$ AVE models which tolerate $\epsilon$ error as:*

$$\mathcal{M}^k(\Pi, \mathcal{V}; \epsilon) = \{\tilde{m} \in \mathcal{M} : \|\tilde{\mathcal{T}}_\pi^k v - \mathcal{T}_\pi^k v\| \leq \epsilon \ \forall \pi \in \Pi, \forall v \in \mathcal{V}\} \tag{5}$$

*where $\|\cdot\|$ denotes the $\ell_\infty$-norm.*

*Moreover, we can analogously define approximate proper value equivalent (APVE) classes as follows:*

$$\mathcal{M}^\infty(\Pi; \epsilon) = \{\tilde{m} \in \mathcal{M} : \|\tilde{v}_\pi - v_\pi\| \leq \epsilon \ \forall \pi \in \Pi\}. \tag{6}$$

### 3.1 Topological properties

In what follows we investigate the topological properties of AVE classes, ultimately showing that, by introducing approximation to VE, we can relate *arbitrary* model classes. An order-$k$ VE class is specified in terms of four quantities: a model class $\mathcal{M}$, a set of policies $\Pi$, a set of functions $\mathcal{V}$, and an order $k$. Grimm et al. [2020] illustrated a variety of topological properties of order-1 VE classes by studying the effect of varying $\mathcal{M}$, $\Pi$ and $\mathcal{V}$. These results were later extended by Grimm et al. [2021], who also investigated the effect of varying the order $k$. We now conduct a similar analysis of AVE classes, which are extensions of VE classes that depend on a fifth quantity: an error tolerance $\epsilon$. We start by showing that AVE models are a strict generalization of VE models, which are a special case when $\epsilon = 0$:

**Property 1.** *For any $\Pi \subseteq \mathbb{\Pi}$, $\mathcal{V} \subseteq \mathbb{V}$ and $\mathcal{M} \subseteq \mathbb{M}$, it follows that $\mathcal{M}^k(\Pi, \mathcal{V}; 0) = \mathcal{M}^k(\Pi, \mathcal{V})$ and $\mathcal{M}^\infty(\Pi; 0) = \mathcal{M}^\infty(\Pi)$.*

Proofs of these properties will be deferred to Appendix A.1. Since Property 1 tells us that AVE is a generalization of exact VE, it is natural to inspect which other topological properties carry over.

**Property 2.** *For any $\epsilon \in \bar{\mathbb{R}}^+$, $\mathcal{M} \subseteq \bar{\mathcal{M}} \subseteq \mathbb{M}$, $\Pi \subseteq \Pi' \subseteq \mathbb{\Pi}$ and $\mathcal{V} \subseteq \mathcal{V}' \subseteq \mathbb{V}$, it follows that*

$$\mathcal{M}^k(\Pi', \mathcal{V}'; \epsilon) \subseteq \mathcal{M}^k(\Pi, \mathcal{V}; \epsilon) \subseteq \bar{\mathcal{M}}^k(\Pi, \mathcal{V}; \epsilon). \tag{7}$$

Property 2 generalizes two of the topological properties described by Grimm et al. [2020] to AVE. In particular, the first subset relation in Eq. 7 shows that, like with VE, as the sizes of the sets of policies and functions increase, the corresponding AVE class shrinks. The second relation shows that as the set of models $\mathcal{M}$ increases in size, so too do AVE classes.

We now investigate the effect of the new parameter, the error tolerance $\epsilon$, on the topology of the resulting AVE classes:

**Property 3.** *For any $\Pi \subseteq \mathbb{\Pi}$, $\mathcal{V} \subseteq \mathbb{V}$ and $\epsilon, \epsilon' \in \bar{\mathbb{R}}^+$ such that $\epsilon' \geq \epsilon$, it follows that*

$$\mathcal{M}^k(\Pi, \mathcal{V}; \epsilon) \subseteq \mathcal{M}^k(\Pi, \mathcal{V}; \epsilon'). \tag{8}$$

Property 3 shows that increasing the tolerated error of an AVE class causes it to include more models. While this is intuitive obvious, (e.g., "a model class that tolerates a higher error also tolerates a lower error"), it may be less straightforward that this property can be used to relate AVE classes with respect to arbitrary orders and sets of functions and policies.

**Proposition 1.** *For any $\epsilon \in \bar{\mathbb{R}}^+$, $\Pi, \Pi' \subseteq \mathbb{\Pi}$, $\mathcal{V}, \mathcal{V}' \subseteq \mathbb{V}$ and $k, K \in \mathbb{Z}^+$ there exists some $\epsilon' \in \bar{\mathbb{R}}^+$ such that*

$$\mathcal{M}^k(\Pi, \mathcal{V}; \epsilon) \subseteq \mathcal{M}^K(\Pi', \mathcal{V}'; \epsilon'). \tag{9}$$

*Moreover, if $\mathcal{M}$, $\mathcal{V}$ and $\mathcal{V}'$ are bounded then $\epsilon'$ is finite.*

Proposition 1 shows that it is possible to relate *arbitrary* AVE classes. This is significantly broader than previous topological results in which VE classes could be related only when their corresponding sets of policies and functions were subsets of each other (when $\Pi \subseteq \Pi'$ or $\mathcal{V} \subseteq \mathcal{V}'$) [Grimm et al., 2020], or when the orders being compared were multiples of each other (when $k$ divided $K$) [Grimm et al., 2021]. The proposition follows directly from taking the limit of $\epsilon' \to \infty$ in Property 3: by tolerating a high enough error *any AVE class can be made to contain any other* (as illustrated in Figure 1).

Why is it important to ensure that one AVE class is a sub-class of another? We know that, if a property applies to a model class, it must also apply to any sub-class. Grimm et al. [2021] have shown that all models in $\mathcal{M}^\infty(\mathbb{\Pi}_{\mathrm{det}}; 0)$ have the same optimal policy as the environment. They also showed that, without further assumptions, this is the "largest" model class with this property. However, prior to our work this was the *only* known theoretical statement about the performance of VE model classes.

Proposition 1 allows us to go much further and provide performance guarantees for *any* AVE class. This is possible because we know we can relate arbitrary AVE classes of the form $\mathcal{M}^k(\Pi, \mathcal{V}; \epsilon)$ to $\mathcal{M}^\infty(\mathbb{\Pi}_{\mathrm{det}}; 0)$ by tolerating a sufficiently large amount of error $\epsilon'$ with respect to the latter model class. Obviously, the tolerated error $\epsilon'$ will play a role in the resulting performance guarantees. In general, the larger this error the weaker the guarantees. So, we are interested in finding the minimum $\epsilon'$ that makes Proposition 1 true.

The minimum $\epsilon'$ for which Proposition 1 holds between any two AVE classes can also be thought of as a function, which we will refer to as the *minimum tolerated error*.

**Definition 2. (Minimum Tolerated Error)** *For any AVE classes $\mathcal{M}^k(\Pi, \mathcal{V}; \epsilon)$ and $\mathcal{M}^K(\Pi', \mathcal{V}'; \epsilon')$ we denote the minimum $\epsilon' \in \bar{\mathbb{R}}^+$ such that $\mathcal{M}^k(\Pi, \mathcal{V}; \epsilon) \subseteq \mathcal{M}^K(\Pi', \mathcal{V}'; \epsilon')$ as $\mathcal{E}_\epsilon(\Pi, \mathcal{V}, k \,|\, \Pi', \mathcal{V}', K)$.*

When $K = \infty$ we omit $\mathcal{V}'$ to reflect the fact that APVE classes do not depend on sets of functions, that is, we use $\mathcal{E}(\Pi, \mathcal{V}, k \,|\, \Pi', \infty)$ (and analogously for $k = \infty$ we omit $\mathcal{V}$). Note that the minimum tolerated error is well defined for any two AVE classes. Equipped with this concept, we can now present the following result:

**Proposition 2.** *For any $\tilde{m} \in \mathcal{M}^k(\Pi, \mathcal{V}; \epsilon)$ it follows that*
$$\|v_{\tilde{\pi}_*} - v_*\| \leq 2 \cdot \mathcal{E}_\epsilon(\Pi, \mathcal{V}, k \,|\, \mathbb{\Pi}, \infty),$$
*where $\tilde{\pi}_*$ is any optimal policy of $\tilde{m}$.*

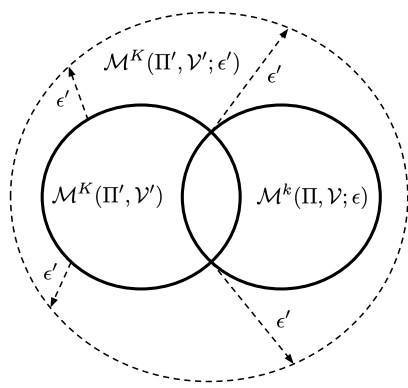

Figure 1: An illustration of the implication of Proposition 1: as $\epsilon'$ increases $\mathcal{M}^K(\Pi', \mathcal{V}'; \epsilon')$ grows— eventually to the point of containing $\mathcal{M}^k(\Pi, \mathcal{V}; \epsilon)$.

Proposition 2 shows that we can we can bound the performance of any AVE class $\mathcal{M}^k(\Pi, \mathcal{V}; \epsilon)$ in terms of its topological relationship with $\mathcal{M}^\infty(\mathbb{\Pi})$ expressed through the minimum error tolerance $\mathcal{E}_\epsilon(\Pi, \mathcal{V}, k \,|\, \mathbb{\Pi}, \infty)$. Accordingly, understanding how this function varies with respect to its inputs is *crucial* to determining the performance of models in $\mathcal{M}^k(\Pi, \mathcal{V}; \epsilon)$. It is worth emphasizing that $\mathcal{E}_\epsilon(\cdot \,|\, \cdot)$ is fundamentally a function concerned with the topology of AVE classes and that Proposition 2 provides a link between such topological properties and the performance of the corresponding models. In the following we will derive a series of novel topological relationships between AVE classes and use them to produce upper bounds on different minimum tolerated errors $\mathcal{E}_\epsilon(\cdot \,|\, \cdot)$. Through combining these relationships and their associated upper bounds we will ultimately be able to derive an upper bound on $\mathcal{E}_\epsilon(\mathbb{\Pi}, \mathcal{V}, k \,|\, \mathbb{\Pi}, \infty)$ which can be used in conjunction with Proposition 2 to provide performance guarantees for a wide range of AVE classes.

# 4 Bounding the minimum tolerated error

Proposition 2 provides a means of converting upper bounds on minimum tolerated errors $\mathcal{E}_\epsilon(\Pi, \mathcal{V}, k \,|\, \mathbb{\Pi}, \infty)$ into performance guarantees for models in $\mathcal{M}^k(\Pi, \mathcal{V}; \epsilon)$. In this section we show how we can use topological relationships between AVE classes to produce such upper bounds.

**Remark 1.** *For any $\epsilon, \epsilon' \in \bar{\mathbb{R}}^+$, $\Pi, \Pi' \subseteq \mathbb{\Pi}$, $\mathcal{V}, \mathcal{V}' \subseteq \mathbb{V}$ and $k, K \in \mathbb{Z}^+$*
$$\mathcal{M}^k(\Pi, \mathcal{V}; \epsilon) \subseteq \mathcal{M}^K(\Pi', \mathcal{V}'; \epsilon') \implies \mathcal{E}_\epsilon(\Pi, \mathcal{V}, k \,|\, \Pi', \mathcal{V}', K) \leq \epsilon'.$$

Remark 1 is a direct consequence of Definition 2, but is useful to state it explicitly since it connects the ensuing topological relationships to the upper bounds on minimum tolerated errors which will be used to construct our performance guarantees. Having established this, we will now develop a series of topological relationships between AVE classes of different kinds and use Remark 1 to generate upper bounds. To facilitate our analysis, we will begin by varying aspects of these AVE classes in isolation, first focusing on those that differ in orders $k$ and $K$ and then those that differ in function sets $\mathcal{V}$ and $\mathcal{V}'$. Later, we combine these results to obtain an upper bound on $\mathcal{E}_\epsilon(\mathbb{\Pi}, \mathcal{V}, k \,|\, \mathbb{\Pi}, \infty)$ which will in turn yield a generic performance bound for models in any AVE class, $\mathcal{M}^k(\mathbb{\Pi}, \mathcal{V}; \epsilon)$, where $k$ and $\mathcal{V}$ are arbitrary orders and function sets.

## 4.1 Model class order and error tolerance

In this section we consider pairs of model classes that share a common set of policies $\Pi$ and functions $\mathcal{V}$, but have different orders $k$ and $K$. In particular, we explore this dependence in the special case in which $k$ divides $K$ and that $\mathcal{V}$ is closed under environment Bellman updates with respect to any of the policies in $\Pi$.

**Proposition 3.** *For any $\epsilon \in \bar{\mathbb{R}}^+$, $\Pi \subseteq \mathbb{\Pi}$, $\mathcal{V} \subseteq \mathbb{V}$ such that $v \in \mathcal{V} \implies \mathcal{T}_\pi v \in \mathcal{V} \, \forall \pi \in \Pi$ and $k, K \in \mathbb{Z}^+$ such that $k$ divides $K$, we have that*
$$\mathcal{M}^k(\Pi, \mathcal{V}; \epsilon) \subseteq \mathcal{M}^K(\Pi, \mathcal{V}; \tfrac{\epsilon \cdot (1 - \gamma^K)}{1 - \gamma^k}). \tag{10}$$

Proposition 3 shows that, in our special case, our upper bound on the minimum tolerated error increases according to a multiplicative factor that depends on the orders $k$ and $K$. For a fixed $k$, as $K$ grows our upper bound will also grow, however the rate of growth is decreasing since the numerator of this factor, $(1 - \gamma^K)$, approaches 1 as $K \to \infty$. The fact that our upper bound does not grow without bound suggests that there is a similar relationship between order-$k$ AVE and APVE classes. Indeed, by sending $K \to \infty$ we obtain such a relationship.

**Corollary 1.** *For any set of policies $\Pi \subseteq \mathbb{\Pi}$, set of functions $\mathcal{V} \in \mathbb{V}$ such that $\{v_\pi : \pi \in \Pi\} \subseteq \mathcal{V}$ and $k \in \mathbb{Z}^+$, it follows that*

$$\mathcal{M}^k(\Pi, \mathcal{V}; \epsilon) \subseteq \mathcal{M}^\infty(\Pi; \tfrac{\epsilon}{1-\gamma^k}). \tag{11}$$

Notice in Corollary 1 how the constraint on $\mathcal{V}$ has been relaxed, since now it is no longer required that $\mathcal{V}$ be closed under arbitrary compositions of the Bellman operators of the policies in $\Pi$. Instead, $\mathcal{V}$ must contain the value functions associated with $\Pi$. This looser constraint implies that $\mathcal{V}$ can be finite when $\Pi$ is finite.

## 4.2 Function sets and error tolerance

In this section we consider two AVE classes with the same order $k$ and set of policies $\Pi$ but different function sets, that is, we compare $\mathcal{M}^k(\Pi, \mathcal{V}; \epsilon)$ and $\mathcal{M}^k(\Pi, \mathcal{V}'; \epsilon')$. Though we are able to provide an upper bound on the minimum tolerated error for arbitrary function sets, it can be shown to scale differently when there are special relationships between $\mathcal{V}$ and $\mathcal{V}'$. We begin by examining this special case.

### 4.2.1 Related function sets

We begin by examining the case in which $\mathcal{V}'$ consists of constrained linear combinations of the elements of $\mathcal{V}$. In particular, we consider a set $\mathcal{V}'$ of the following form:

$$c\text{-vspan}(\mathcal{V}) \equiv \left\{ \sum_i \alpha_i v_i(s) : \ \forall \alpha_i \quad \sum_i \alpha_i = 1, \sum_i |\alpha_i| \leq c \right\}, \tag{12}$$

which we refer to as the *constrained vspan of $\mathcal{V}$ with parameter $c$.* The coefficients of the linear combinations in $c$-vspan$(\mathcal{V})$ are constrained to sum to 1 and to have magnitudes which sum to less than $c$.

Grimm et al. [2020] studied a special case of Eq. 12 without the coefficient magnitude constraint (by taking $c \to \infty$) and showed that $\mathcal{M}^1(\Pi, \mathcal{V}; 0) = \mathcal{M}^1(\Pi, \mathcal{V}'; 0)$ (where $\mathcal{V}' = \infty$-vspan$(\mathcal{V})$). Notice how, in the language of Proposition 1, we have just related two VE classes with different sets of functions $\mathcal{V}$ and $\mathcal{V}'$ without tolerating any additional error. However, this result is relatively narrow, only applying to exact order-1 VE classes. In what follows we extend this relationship to order-$k$ AVE classes:

**Proposition 4.** *For any set of policies $\Pi \subseteq \mathbb{\Pi}$, set of functions $\mathcal{V} \in \mathbb{V}$, $c > 1$ and error $\epsilon \in \bar{\mathbb{R}}^+$, we have*

$$\mathcal{M}^k(\Pi, c\text{-vspan}(\mathcal{V}); \epsilon) \subseteq \mathcal{M}^k(\Pi, \mathcal{V}; \epsilon) \subseteq \mathcal{M}^k(\Pi, c\text{-vspan}(\mathcal{V}); c \cdot \epsilon). \tag{13}$$

Proposition 4 shows how $\mathcal{M}^k(\Pi, \mathcal{V}; \epsilon)$ is "squeezed" between two other model classes whose function sets are replaced by $c$-vspan$(\mathcal{V})$. In particular, by considering the rightmost subset relationship in Eq. 13, we see that the error term scales multiplicatively with $c$. It is interesting to note that, for particular choices of $\epsilon$ and $c$, the left- and right-most terms in Eq. 13 can be made identical, reducing the expression to an equality.

**Corollary 2.** *When either $c = 1$ or $\epsilon = 0$, for any $\Pi \subseteq \mathbb{\Pi}$, $\mathcal{V} \subseteq \mathbb{V}$ it follows that*

$$\mathcal{M}^k(\Pi, \mathcal{V}; \epsilon) = \mathcal{M}^k(\Pi, c\text{-vspan}(\mathcal{V}); \epsilon). \tag{14}$$

Note that, when $\epsilon = 0$, Corollary 2 recovers Grimm et al. [2020]'s result by sending $c \to \infty$. On the other hand, when $c = 1$, $c$-vspan$(\mathcal{V})$ reduces to the convex hull of $\mathcal{V}$. In the context of Proposition 1, this means that, regardless of $\epsilon$ and $k$, we do not need to tolerate any additional error for $\mathcal{M}^k(\Pi, 1\text{-vspan}(\mathcal{V}); \epsilon)$ to contain $\mathcal{M}^k(\Pi, \mathcal{V}; \epsilon)$.

### 4.2.2 Unrelated function sets

In the previous section we studied the case in which the set $\mathcal{V}'$ coincided with $c\text{-vspan}(\mathcal{V})$; we now consider a more general scenario where $\mathcal{V}$ and $\mathcal{V}'$ are not assumed to have any relationship to each other. In this setting, we will measure the similarity of sets of functions using a distance function $\beta : 2^{\mathbb{V}} \times 2^{\mathbb{V}} \to \mathbb{R}$ defined as

$$\beta(\mathcal{V}||\mathcal{V}') \equiv \max_{v' \in \mathcal{V}'} \min_{v \in \mathcal{V}} \|v - v'\|. \tag{15}$$

In words, $\beta$ measures the greatest distance from any element in $\mathcal{V}'$ to its closest element in $\mathcal{V}$. Before we continue, it is worth pointing out some interesting properties of $\beta$.

**Proposition 5.**

1. *(**Asymmetry**) For any $\mathcal{V} \subseteq \mathcal{V}' \subseteq \mathcal{V}'' \subseteq \mathbb{V}$ it follows that*

$$0 = \beta(\mathcal{V}||\mathcal{V}) \leq \beta(\mathcal{V}||\mathcal{V}') \leq \beta(\mathcal{V}||\mathcal{V}'') \quad and \quad 0 = \beta(\mathcal{V}''||\mathcal{V}'') \leq \beta(\mathcal{V}'||\mathcal{V}'') \leq \beta(\mathcal{V}||\mathcal{V}'').$$

2. *(**Convex, Compact $\mathcal{V}$**) When $\mathcal{V}$ is convex and compact it follows that*

$$\beta(\mathcal{V}||\mathcal{V}') = \beta(\mathcal{V}||1\text{-vspan}(\mathcal{V}')).$$

Proposition 5 illustrates several interesting features of $\beta$. Firstly, it measures the similarity between sets $\mathcal{V}$ and $\mathcal{V}'$ asymmetrically: if $\mathcal{V}'$ grows then $\beta$ also increases, since there are more elements to maximize over; however, if $\mathcal{V}$ grows, $\beta$ *decreases* instead, since there are more elements to minimize over. Secondly, when $\mathcal{V}$ is a convex compact set, $\mathcal{V}'$ can be replaced with its convex-hull (equivalently, $1\text{-vspan}(\cdot)$) without changing $\beta$.

Using the similarity measure $\beta$, we can then determine the amount of error an AVE class with respect to $\mathcal{V}'$ must tolerate to contain a counterpart defined with respect to $\mathcal{V}$.

**Proposition 6.** *For any $\Pi \in \mathbb{\Pi}$, $\mathcal{V}, \mathcal{V}' \in \mathbb{V}$ and $\epsilon \in \bar{\mathbb{R}}^+$, it follows that*

$$\mathcal{M}^k(\Pi, \mathcal{V}; \epsilon) \subseteq \mathcal{M}^k(\Pi, \mathcal{V}'; \epsilon + 2\gamma^k \beta(\mathcal{V}||\mathcal{V}')),$$

*moreover, if $\mathcal{V}$ is convex and compact, we obtain:*

$$\mathcal{M}^k(\Pi, \mathcal{V}; \epsilon) \subseteq \mathcal{M}^k(\Pi, 1\text{-vspan}(\mathcal{V}'); \epsilon + 2\gamma^k \beta(\mathcal{V}||\mathcal{V}')).$$

Proposition 6 shows that, for any two sets of functions $\mathcal{V}$ and $\mathcal{V}'$, our upper bound on the minimum tolerated error is proportional to $\beta(\mathcal{V}||\mathcal{V}')$, which we analyzed in Proposition 5. In particular, we know that our upper bound increases as $\mathcal{V}'$ grows and decreases as $\mathcal{V}$ grows. In addition, in the special case in which $\mathcal{V}$ is a convex compact set, $\mathcal{V}'$ can be replaced by its convex hull without any increase in our upper bound.

### 4.3 Unifying results

In the previous sections we studied upper bounds on the minimum tolerated error such that models in particular kinds of AVE classes would be contained in other kinds of AVE classes. Each section concerned a different type of upper bound constructed by varying certain aspects of the AVE classes: Section 4.1 held function sets fixed and varied model-class orders whereas Section 4.2 held model-class orders fixed and varied function sets. Now, we combine these results to obtain a more general theorem.

**Theorem 1.** *For any $\Pi \subseteq \mathbb{\Pi}$, $\mathcal{V} \subseteq \mathbb{V}$, $k \in \mathbb{Z}^+$, and $\epsilon \in \bar{\mathbb{R}}^+$ it follows that*

$$\mathcal{M}^k(\Pi, \mathcal{V}; \epsilon) \subseteq \mathcal{M}^\infty(\Pi; \tfrac{1}{1-\gamma^k} \cdot \min_{c \geq 1}(c \cdot \epsilon + 2\gamma^k \beta(c\text{-vspan}(\mathcal{V})||\mathbb{V}_\Pi)))$$

*which implies that*

$$\mathcal{E}_\epsilon(\Pi, \mathcal{V}, k \,|\, \Pi, \infty) \leq \tfrac{1}{1-\gamma^k} \cdot \min_{c \geq 1}(c \cdot \epsilon + 2\gamma^k \beta(c\text{-vspan}(\mathcal{V})||\mathbb{V}_\Pi)).$$

*Proof.* The proof follows by a sequential application of our previous results.

$$\mathcal{M}^k(\Pi, \mathcal{V}; \epsilon) \subseteq \mathcal{M}^k(\Pi, c\text{-vspan}(\mathcal{V}); c \cdot \epsilon) \qquad \text{(Proposition 4)}$$

$$\subseteq \mathcal{M}^k(\Pi, \mathbb{V}_\Pi; c \cdot \epsilon + 2\gamma^k \beta(c\text{-vspan}(\mathcal{V})||\mathbb{V}_\Pi)) \qquad \text{(Proposition 6)}$$

$$= \bigcap_{\pi \in \Pi} \bigcap_{v_{\pi'} \in \mathbb{V}_\Pi} \mathcal{M}^k(\{\pi\}, \{v_{\pi'}\}; c \cdot \epsilon + 2\gamma^k \beta(c\text{-vspan}(\mathcal{V})||\mathbb{V}_\Pi))$$

$$\subseteq \bigcap_{\pi \in \Pi} \mathcal{M}^k(\{\pi\}, \{v_\pi\}; c \cdot \epsilon + 2\gamma^k \beta(c\text{-vspan}(\mathcal{V})||\mathbb{V}_\Pi)) \qquad (16)$$

$$\subseteq \bigcap_{\pi \in \Pi} \mathcal{M}^\infty(\{\pi\}; \tfrac{1}{1-\gamma^k}(c \cdot \epsilon + 2\gamma^k \beta(c\text{-vspan}(\mathcal{V})||\mathbb{V}_\Pi))) \qquad \text{(Corollary 1)}$$

$$= \mathcal{M}^\infty(\Pi; \tfrac{1}{1-\gamma^k}(c \cdot \epsilon + 2\gamma^k \beta(c\text{-vspan}(\mathcal{V})||\mathbb{V}_\Pi))).$$

Since Eq. 16 holds for all $c \geq 1$, it follows that:

$$\mathcal{M}^k(\Pi, \mathcal{V}; \epsilon) \subseteq \mathcal{M}^\infty(\Pi; \tfrac{1}{1-\gamma^k} \cdot \min_{c \geq 1}(c \cdot \epsilon + 2\gamma^k \beta(c\text{-vspan}(\mathcal{V})||\mathbb{V}_\Pi))).$$

Finally, using Remark 1 we obtain the associated upper bound on $\mathcal{E}_\epsilon(\Pi, \mathcal{V}, k \,|\, \Pi, \infty)$. $\qquad \square$

Theorem 1 reveals an interesting tension. Notice how the quantity inside the minimum is determined by two terms, $c \cdot \epsilon$ and $2\gamma^k \beta(c\text{-vspan}(\mathcal{V})||\mathcal{V}')$. As $c$ grows, the first term increases; however, due to the properties of $\beta$ described in Proposition 5, the second term will shrink.

## 5    Bounds on model performance

Taken together Propositions 1 and 2 and Remark 1 provide a pipeline for quantifying the performance for *any* AVE class $\mathcal{M}^k(\Pi, \mathcal{V}; \epsilon)$. Propositions 1 and 2 form a powerful connection between the general topology of AVE classes and model performance in terms of minimum tolerated errors (Definition 2). Remark 1 then allows us to use *specific* topological relationships between kinds of AVE classes to quantify these minimum tolerated errors using upper bounds (see Section 4). Accordingly, in Section 4 we produced topological results about AVE models when policy sets $\Pi$ and either order or function sets are held fixed. We ultimately combined these results into Theorem 1 resulting in a bound on the minimum tolerated error $\mathcal{E}_\epsilon(\Pi, \mathcal{V}, k \,|\, \Pi, \infty)$.

By combining Theorem 1 with Proposition 2 we can obtain a performance bound on AVE classes with respect to arbitrary orders and function sets:

**Theorem 2.** *For any $\tilde{m} \in \mathcal{M}^k(\Pi, \mathcal{V}; \epsilon)$ it follows that*

$$\|v_* - v_{\tilde{\pi}_*}\| \leq \tfrac{2}{1-\gamma^k} \cdot \min_{c \geq 1}\left(c \cdot \epsilon + 2\gamma^k \beta(c\text{-vspan}(\mathcal{V})||\mathbb{V}_\Pi)\right), \qquad (17)$$

*where $\tilde{\pi}_*$ is an optimal policy of $\tilde{m}$.*

Theorem 2 suggests that ideal AVE classes have low error $\epsilon$ and function sets such that $c\text{-vspan}(\mathcal{V}) \supseteq \mathbb{V}_\Pi$. However, in practice an agent does not directly have control over the error $\epsilon$ with which it learns a model. Rather, this error is dependent on the choice of $\mathcal{V}$: some functions may be harder to learn VE models with respect to than others. Since finding a model that is VE with respect to a small number of functions is generally easier, this dependence between $\epsilon$ and $\mathcal{V}$ introduces a tension in the selection of the set $\mathcal{V}$. The smaller the number of functions in $\mathcal{V}$, the smaller the resulting $\epsilon$ tends to be, but on the other hand the distance $\beta(c\text{-vspan}(\mathcal{V})||\mathbb{V}_\Pi)$ generally increases (see Section 6 for an empirical illustration of this tension).

One option for choosing $\mathcal{V}$, which was explored in both Grimm et al. [2020] and Grimm et al. [2021], is to use some manageable set of value functions: $\{v_{\pi_i}\}_{i=1}^n$. This is sensible as it is known that value functions form a polytope $\mathbb{V}_\Pi$ which could be covered by $c\text{-vspan}(\cdot)$ [Dadashi et al., 2019]. However, value functions are quantities of the environment which are not assumed to be known by the agent. At best, an agent can produce *estimates* of such quantities.

Interestingly, our AVE formalism can address not only the error in the satisfaction of VE constraints but also in the estimation of functions that the constraints depend upon. If an agent can estimate each $v_{\pi_i}$ by $\hat{v}_{\pi_i}$ with bounded approximation error $\|v_{\pi_i} - \hat{v}_{\pi_i}\| \leq \epsilon_{\text{approx}}$, we have:

$$\mathcal{M}^k(\Pi, \{\hat{v}_{\pi_i}\}_{i=1}^n; \epsilon) \subseteq \mathcal{M}^k(\Pi, \{v_{\pi_i}\}_{i=1}^n; \epsilon + 2\gamma^k \beta(\{\hat{v}_{\pi_i}\}_{i=1}^n \| \{v_{\pi_i}\}_{i=1}^n))$$
$$\subseteq \mathcal{M}^k(\Pi, \{v_{\pi_i}\}_{i=1}^n; \epsilon + 2\gamma^k \epsilon_{\text{approx}}). \tag{18}$$

Eq. 18 shows that we can *exchange* the estimation error of functions in VE function sets for error in the satisfaction of VE constraints. That is, any model in the AVE class with respect to approximate value functions is also AVE with respect to actual value functions if we tolerate additional error proportional to their estimation error. This ultimately means that we can bound the performance of models which are approximately VE with respect to functions that are not in $\mathbb{V}_\Pi$ (that is, that are not value functions).

**Corollary 3.** *Let $\hat{\mathbb{V}}_\Pi = \{\hat{v}_\pi : \pi \in \Pi\}$ be a set of approximate value functions satisfying $\|v_\pi - \hat{v}_\pi\| \leq \epsilon_{approx}$ for all $\pi \in \Pi$. Then for any $\tilde{m} \in \mathcal{M}^k(\Pi, \hat{\mathbb{V}}_\Pi; \epsilon)$ it follows that:*

$$\|v_* - v_{\tilde{\pi}_*}\| \leq \frac{2(\epsilon + 2\gamma^k \epsilon_{approx})}{1 - \gamma^k},$$

*where $\tilde{\pi}_*$ is any optimal policy in $\tilde{m}$.*

## 6 Trade-offs between model capacity and function set size

In this section we show that our approximate formulation of VE can guide the choice of functions to learn a VE model with respect to when model capacity is limited. In particular, we show that, depending on the situation, an agent may prefer to either learn a VE model with respect to a larger class of functions (and tolerate a higher approximation error $\epsilon$) or learn a VE model with respect to a smaller class of functions (and tolerate a lower $\epsilon$). These situations are interesting because, as we showed in Theorem 2, higher approximation error results in worse bounds on the planning performance of the associated models—which raises the possibility that in certain situations a model with limited capacity might plan better when it is VE with respect to fewer functions.

To illustrate situations where this might occur, we consider the tabular Four Rooms domain [Sutton et al., 1999] and learn tabular VE models whose per-action transition matrices are constrained to have rank at most $R$. We learn these models to be in the VE class $\mathcal{M}(\Pi, \mathcal{V})$, where $\mathcal{V}$ is a set of $D$ functions generated by sampling $v(s) \sim \text{Uniform}(-10, 10)$ for each $v \in \mathcal{V}$ and each $s \in \mathcal{S}$. Due to the rank constraint on the model, we will only be able to learn approximate VE models with varying error tolerances for different settings of $R$ and $D$.

Figure 2 shows a histogram of the planning performance of such models. Each cell in the figure corresponds to a model associated with specific values of $R$ and $D$, and the cells' color denotes the value of the model's optimal policy averaged over states and over 10 independent executions. Interestingly, near the bottom left portion of the histogram there is a region in which low-capacity VE models with respect to small sets of functions perform better than their counterparts generated using more functions.

## 7 Related work

Our work directly extends previous theoretical results on VE to the approximate setting. This naturally situates us closest to the works that originally explicated value equivalence and its higher order analogues [Grimm et al., 2020, 2021]. Our effort in deriving performance guarantees for VE is also reminiscent of classical works that studied approximate forms of policy and value iteration [Bertsekas and Tsitsiklis, 1995, Munos, 2005].

A closely related line of work is value-aware model learning (VAML, IterVAML, VaGraM) [Farahmand et al., 2017, Farahmand, 2018, Voelcker et al., 2021]). In VAML models are learned in order to minimize the discrepancy between their Bellman optimality operators and those of the environment. In practice, a family of functions is constructed and a model is learned to minimize the maximum discrepancy across this family. In IterVAML, instead of minimizing the worst case discrepancy across a family of functions, the functions are generated by the model during a value iteration procedure.

In VaGraM, the authors consider a similar discrepancy-based loss as in VAML, but use a Taylor series approximation to avoid having loss terms involving the values of predicted future states. This ultimately gives rise to a squared-error loss between predicted and actual next states weighted by the gradients of a value function. VE and VAML can be seen as complementary: both concern models that are learned with their eventual use in mind, but VAML focuses on the associated optimization problems of finding such models whereas VE focuses on characterizing these models into classes (and also on providing performance guarantees in this work).

There have also been several recent empirical works which explored concepts adjacent to VE. Farquhar et al. [2021] explored adding auxiliary terms to different model learning procedures which encouraged the learned models' Bellman operators and value estimates to satisfy the Bellman equations (i.e., to be self-consistent). They show that there are circumstances in which this leads to better data-efficiency in practice. Another example is Nikishin et al. [2021], who study the problem of learning models whose Bellman optimality operators induce optimal policies in the environment. In the service of this, they propose using implicit differentiation [Christianson, 1994] to compute gradients from the model's optimality operator all the way to its optimal policy.

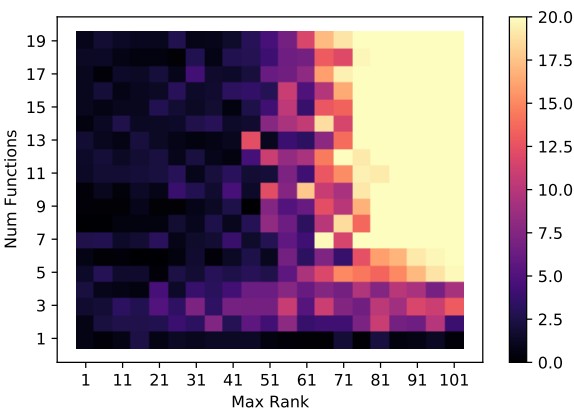

Figure 2: Illustration of the trade-offs between VE model capacity and the size of $\mathcal{V}$. From left-to-right the capacity of the models increase, from bottom-to-top the size of $\mathcal{V}$ increases. Each cell is colored according to the average value of the model's optimal policy.

# 8 Conclusion and future work

We provided an approximate theory of value equivalence that extends the existing VE formalism [Grimm et al., 2020, 2021]. The main concept introduced was a family of AVE classes whose models can violate VE constraints up to a limited degree. We showed that, as a product of introducing approximation to VE, arbitrary AVE classes can be related to each other topologically. Moreover, we showed that, by relating them to specific VE classes for which there are performance guarantees, we can derive bounds on the performance of arbitrary AVE classes in terms of their topological properties. Motivated by this link between topology and performance bounds, we proceed to derive isolated topological relationships between AVE classes whose function sets, orders and error tolerances vary, which we then can combine to produce bounds which hold over AVE classes with respect to arbitrary orders, functions sets and error tolerances. This addresses a fundamental open question in the VE theory: how to provide performance guarantees for models that do not satisfy the VE constraints exactly. It may be possible to derive topological relationships considering all constraints of a VE class jointly and, in so doing, provide tighter performance bounds, but we leave this for future work. We also believe that the performance bounds we have introduced might give rise to new algorithms that attempt to minimize them, either directly or indirectly.

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
