# A   Appendix

## A.1   Proofs

In this section we restate and provide proofs of the statements made in the main text.

**Property 1.** *For any* $\Pi \subseteq \mathbb{\Pi}$, $\mathcal{V} \subseteq \mathbb{V}$ *and* $\mathcal{M} \subseteq \mathbb{M}$, *it follows that* $\mathcal{M}^k(\Pi, \mathcal{V}; 0) = \mathcal{M}^k(\Pi, \mathcal{V})$ *and* $\mathcal{M}^\infty(\Pi; 0) = \mathcal{M}^\infty(\Pi)$.

*Proof.* Any $\tilde{m} \in \mathcal{M}^k(\Pi, \mathcal{V}; 0)$ satisfies $\|\tilde{\mathcal{T}}_\pi^k v - \mathcal{T}_\pi^k v\| = 0 \ \forall \pi \in \Pi, \forall v \in \mathcal{V}$. Similarly any $\tilde{m} \in \mathcal{M}^k(\Pi, \mathcal{V})$ satisfies analogous equality constraints $\tilde{\mathcal{T}}_\pi^k v = \mathcal{T}_\pi^k v \ \forall \pi \in \Pi, \forall v \in \mathcal{V}$. Since $\| \cdot \|$ is a norm, we know that $\|\tilde{\mathcal{T}}_\pi^k v - \mathcal{T}_\pi^k v\| = 0 \iff \tilde{\mathcal{T}}_\pi^k v = \mathcal{T}_\pi^k v$, hence $\mathcal{M}^k(\Pi, \mathcal{V}; 0) = \mathcal{M}^k(\Pi, \mathcal{V})$. The same logic applies to APVE classes. $\qquad\square$

**Property 2.** *For any* $\epsilon \in \bar{\mathbb{R}}^+$, $\mathcal{M} \subseteq \bar{\mathcal{M}} \subseteq \mathbb{M}$, $\Pi \subseteq \Pi' \subseteq \mathbb{\Pi}$ *and* $\mathcal{V} \subseteq \mathcal{V}' \subseteq \mathbb{V}$, *it follows that*

$$\mathcal{M}^k(\Pi', \mathcal{V}'; \epsilon) \subseteq \mathcal{M}^k(\Pi, \mathcal{V}; \epsilon) \subseteq \bar{\mathcal{M}}^k(\Pi, \mathcal{V}; \epsilon). \tag{7}$$

*Proof.* An AVE class, $\mathcal{M}^k(\Pi', \mathcal{V}'; \epsilon)$ satisfies a series of constraints of the form $\|\tilde{\mathcal{T}}_\pi^k v - \mathcal{T}_\pi^k v\| \le \epsilon$ for each pair of $\pi, v \in \Pi' \times \mathcal{V}'$. Considering another pair of sub-sets $\Pi \subseteq \Pi'$ and $\mathcal{V} \subseteq \mathcal{V}'$, we can partition the first pair as follows:

$$\Pi' \times \mathcal{V}' = (\Pi' \setminus \Pi \times \mathcal{V}' \setminus \mathcal{V}) \uplus (\Pi' \setminus \Pi \times \mathcal{V}) \uplus (\Pi \times \mathcal{V}' \setminus \mathcal{V}) \uplus (\Pi \times \mathcal{V})$$

accordingly,

$$\mathcal{M}^k(\Pi', \mathcal{V}'; \epsilon) = \mathcal{M}^k(\Pi' \setminus \Pi, \mathcal{V}' \setminus \mathcal{V}; \epsilon) \cap \mathcal{M}^k(\Pi' \setminus \Pi, \mathcal{V}; \epsilon) \cap \mathcal{M}^k(\Pi, \mathcal{V}' \setminus \mathcal{V}; \epsilon) \cap \mathcal{M}^k(\Pi, \mathcal{V}; \epsilon)$$
$$\subseteq \mathcal{M}^k(\Pi, \mathcal{V}; \epsilon),$$

satisfying the first subset relation in Eq. 7. For the next subset relation, we simply note that

$$\bar{\mathcal{M}}^k(\Pi, \mathcal{V}; \epsilon) = (\bar{\mathcal{M}} \setminus \mathcal{M})^k(\Pi, \mathcal{V}; \epsilon) \cup \mathcal{M}^k(\Pi, \mathcal{V}; \epsilon) \supseteq \mathcal{M}^k(\Pi, \mathcal{V}; \epsilon),$$

completing the proof. $\qquad\square$

**Property 3.** *For any* $\Pi \subseteq \mathbb{\Pi}$, $\mathcal{V} \subseteq \mathbb{V}$ *and* $\epsilon, \epsilon' \in \bar{\mathbb{R}}^+$ *such that* $\epsilon' \ge \epsilon$, *it follows that*

$$\mathcal{M}^k(\Pi, \mathcal{V}; \epsilon) \subseteq \mathcal{M}^k(\Pi, \mathcal{V}; \epsilon'). \tag{8}$$

*Proof.* For any $\tilde{m} \in \mathcal{M}^k(\Pi, \mathcal{V}; \epsilon)$ a number of AVE constraints are respected: $\|\tilde{\mathcal{T}}_\pi^k v - \mathcal{T}_\pi^k v\| \le \epsilon$ for each pair $\pi, v \in \Pi \times \mathcal{V}$. Since $\epsilon' \ge \epsilon$, it follows that $\|\tilde{\mathcal{T}}_\pi^k v - \mathcal{T}_\pi^k v\| \le \epsilon \le \epsilon'$ as well and hence $\tilde{m} \in \mathcal{M}^k(\Pi, \mathcal{V}; \epsilon')$. Thus $\mathcal{M}^k(\Pi, \mathcal{V}; \epsilon) \subseteq \mathcal{M}^k(\Pi, \mathcal{V}; \epsilon')$ as needed. $\qquad\square$

**Proposition 1.** *For any* $\epsilon \in \bar{\mathbb{R}}^+$, $\Pi, \Pi' \subseteq \mathbb{\Pi}$, $\mathcal{V}, \mathcal{V}' \subseteq \mathbb{V}$ *and* $k, K \in \mathbb{Z}^+$ *there exists some* $\epsilon' \in \bar{\mathbb{R}}^+$ *such that*

$$\mathcal{M}^k(\Pi, \mathcal{V}; \epsilon) \subseteq \mathcal{M}^K(\Pi', \mathcal{V}'; \epsilon'). \tag{9}$$

*Moreover, if* $\mathcal{M}$, $\mathcal{V}$ *and* $\mathcal{V}'$ *are bounded then* $\epsilon'$ *is finite.*

*Proof.* Denote $v_{\max} = \max_{s \in \mathcal{S}, v \in \mathcal{V} \cup \mathcal{V}'} v(s)$, $\tilde{r}_{\max} = \max_{s \in \mathcal{S}, a \in \mathcal{A}, \tilde{m} \in \mathcal{M}} r(s, a)$ and consider any $\tilde{m} \in \mathcal{M}^k(\Pi, \mathcal{V}; \epsilon)$. We can then write

$$\|\tilde{\mathcal{T}}_\pi^K v - \mathcal{T}_\pi^K v\| \le \max_s |\tilde{\mathcal{T}}_\pi^K v(s)| + \max_s |\mathcal{T}_\pi^K v(s)| \le 2 \max\{\tilde{r}_{\max}, r_{\max}\} \frac{1 - \gamma^K}{1 - \gamma} + \gamma^K v_{\max}$$

for any $\pi \in \Pi'$, $v \in \mathcal{V}'$ and $\tilde{m} \in \mathcal{M}^k(\Pi, \mathcal{V}; \epsilon)$.

Clearly, when $\epsilon' = \infty$ the desired subset relation holds, as $\mathcal{M}^K(\Pi', \mathcal{V}'; \infty) = \mathcal{M} \supseteq \mathcal{M}^k(\Pi, \mathcal{V}; \epsilon)$ for any choices of sets, orders and $\epsilon$. Additionally, when $\mathcal{M}$, $\mathcal{V}$ and $\mathcal{V}'$ are bounded, we know that $\tilde{r}_{\max}$ and $v_{\max}$ are finite. Thus, by selecting a finite $\epsilon' > 2 \max\{\tilde{r}_{\max}, r_{\max}\} \frac{1 - \gamma^K}{1 - \gamma} + \gamma^K v_{\max}$, we obtain $\tilde{m} \in \mathcal{M}^K(\Pi', \mathcal{V}'; \epsilon')$ and thus $\mathcal{M}^k(\Pi, \mathcal{V}; \epsilon) \subseteq \mathcal{M}^K(\Pi', \mathcal{V}'; \epsilon')$ as needed.

$\qquad\square$

**Proposition 2.** *For any $\tilde{m} \in \mathcal{M}^k(\Pi, \mathcal{V}; \epsilon)$ it follows that*

$$\|v_{\tilde{\pi}_*} - v_*\| \leq 2 \cdot \mathcal{E}_\epsilon(\Pi, \mathcal{V}, k \,|\, \Pi, \infty),$$

*where $\tilde{\pi}_*$ is any optimal policy of $\tilde{m}$.*

*Proof.* From Proposition 1, we know that a minimum tolerated error, $\epsilon' = \mathcal{E}_\epsilon(\Pi, \mathcal{V}, k \,|\, \Pi, \infty)$, exists such that $\mathcal{M}^k(\Pi, \mathcal{V}; \epsilon) \subseteq \mathcal{M}^\infty(\Pi; \epsilon')$. We can then consider the performance of models in $\mathcal{M}^\infty(\Pi; \epsilon')$. For any $\tilde{m} \in \mathcal{M}^\infty(\Pi; \epsilon')$ we can write:

$$
\begin{aligned}
0 \geq \tilde{v}_{\pi_*}(s) - \tilde{v}_{\tilde{\pi}_*}(s) \\
= (\tilde{v}_{\pi_*}(s) - v_{\pi_*}(s)) + (v_{\pi_*} - v_{\tilde{\pi}_*}(s)) + (v_{\tilde{\pi}_*}(s) - \tilde{v}_{\tilde{\pi}_*}(s))
\end{aligned}
\tag{19}
$$

for any $s \in \mathcal{S}$ where $\pi_*$ and $\tilde{\pi}_*$ are arbitrary optimal policies in the environment and $\tilde{m}$ respectively and $\tilde{v}_\pi$ denotes the model's value of a policy $\pi$.

Since $\tilde{m} \in \mathcal{M}^\infty(\Pi; \epsilon')$ we know the first and third terms are bounded below by $-\epsilon'$, giving:

$$
\begin{aligned}
0 &\geq v_*(s) - v_{\tilde{\pi}_*}(s) - 2\epsilon' \\
\implies 2\epsilon' &\geq v_*(s) - v_{\tilde{\pi}_*}(s) \geq 0 \\
\implies \|v_* &- v_{\tilde{\pi}_*}\| \leq 2\epsilon',
\end{aligned}
\tag{20}
$$

as needed. $\qquad\square$

**Proposition 3.** *For any $\epsilon \in \bar{\mathbb{R}}^+$, $\Pi \subseteq \Pi$, $\mathcal{V} \subseteq \mathbb{V}$ such that $v \in \mathcal{V} \implies \mathcal{T}_\pi v \in \mathcal{V} \,\forall \pi \in \Pi$ and $k, K \in \mathbb{Z}^+$ such that $k$ divides $K$, we have that*

$$\mathcal{M}^k(\Pi, \mathcal{V}; \epsilon) \subseteq \mathcal{M}^K(\Pi, \mathcal{V}; \tfrac{\epsilon \cdot (1-\gamma^K)}{1-\gamma^k}).\tag{10}$$

*Proof.* Let $K = nk$, and consider a model $\tilde{m} \in \mathcal{M}^k(\Pi, \mathcal{V}; \epsilon)$. It follows for any $\pi \in \Pi$ and $v \in \mathcal{V}$ that

$$
\begin{aligned}
\|\tilde{\mathcal{T}}_\pi^K v - \mathcal{T}_\pi^K v\| &= \|\tilde{\mathcal{T}}_\pi^k \tilde{\mathcal{T}}_\pi^{K-k} v - \mathcal{T}_\pi^k \mathcal{T}_\pi^{K-k} v\| \\
&= \|\tilde{\mathcal{T}}_\pi^k \tilde{\mathcal{T}}_\pi^{K-k} v - \mathcal{T}_\pi^k \mathcal{T}_\pi^{K-k} v + \tilde{\mathcal{T}}_\pi^k \mathcal{T}_\pi^{K-k} v - \tilde{\mathcal{T}}_\pi^k \mathcal{T}_\pi^{K-k} v\| \\
&\leq \|\tilde{\mathcal{T}}_\pi^k \mathcal{T}_\pi^{K-k} v - \mathcal{T}_\pi^k \mathcal{T}_\pi^{K-k} v\| + \|\tilde{\mathcal{T}}_\pi^k \tilde{\mathcal{T}}_\pi^{K-k} v - \tilde{\mathcal{T}}_\pi^k \mathcal{T}_\pi^{K-k} v\| \\
&\overset{(1)}{\leq} \epsilon + \|\tilde{\mathcal{T}}_\pi^k \tilde{\mathcal{T}}_\pi^{K-k} v - \tilde{\mathcal{T}}_\pi^k \mathcal{T}_\pi^{K-k} v\| \\
&\overset{(2)}{\leq} \epsilon + \gamma^k \|\tilde{\mathcal{T}}_\pi^{K-k} v - \mathcal{T}_\pi^{K-k} v\|
\end{aligned}
\tag{21}
$$

where (1) follows from the assumption on $\mathcal{V}$ and (2) follows from the fact that $\tilde{\mathcal{T}}_\pi$ is a contraction. Next, using induction we can say that:

$$
\begin{aligned}
\|\tilde{\mathcal{T}}_\pi^K v_\pi - \mathcal{T}_\pi^K v_\pi\| &\leq \epsilon \cdot \left(1 + \gamma^k + \gamma^{2k} + \cdots + \gamma^{(n-1)k}\right) \\
&= \epsilon \cdot \sum_{t=0}^{n-1} \gamma^{kt} \\
&= \epsilon \cdot \frac{1-(\gamma^k)^n}{1-\gamma^k} \\
&= \epsilon \cdot \frac{1-\gamma^K}{1-\gamma^k}
\end{aligned}
\tag{22}
$$

where the last equality follows because $K = nk$.

This suffices to show that $\tilde{m} \in \mathcal{M}^K(\Pi, \mathcal{V}; \epsilon \cdot \frac{1-\gamma^K}{1-\gamma^k})$ and thus: $\mathcal{M}^k(\Pi, \mathcal{V}; \epsilon) \subseteq \mathcal{M}^K(\Pi, \mathcal{V}; \epsilon \cdot \frac{1-\gamma^K}{1-\gamma^k})$ as needed. $\qquad\square$

**Corollary 1.** *For any set of policies $\Pi \subseteq \Pi$, set of functions $\mathcal{V} \in \mathbb{V}$ such that $\{v_\pi : \pi \in \Pi\} \subseteq \mathcal{V}$ and $k \in \mathbb{Z}^+$, it follows that*

$$\mathcal{M}^k(\Pi, \mathcal{V}; \epsilon) \subseteq \mathcal{M}^\infty(\Pi; \tfrac{\epsilon}{1-\gamma^k}).\tag{11}$$

*Proof.*

$$\mathcal{M}^k(\Pi, \mathcal{V}; \epsilon) = \bigcap_{\pi \in \Pi} \bigcap_{v \in \mathcal{V}} \mathcal{M}^k(\{\pi\}, \{v\}; \epsilon) \subseteq \bigcap_{\pi \in \Pi} \mathcal{M}^k(\{\pi\}, \{v_\pi\}; \epsilon) \tag{23}$$

where the subset-relation holds from our assumption that $\{v_\pi : \pi \in \Pi\} \subseteq \mathcal{V}$.

Next we examine $\tilde{m} \in \mathcal{M}^k(\{\pi\}, \{v_\pi\}; \epsilon)$ for individual $\pi \in \Pi$. We know that for any such model:

$$\|\tilde{\mathcal{T}}_\pi^{nk} v_\pi - v_\pi\| \leq \|\tilde{\mathcal{T}}_\pi^{nk} v_\pi - \tilde{\mathcal{T}}_\pi^k v_\pi\| + \|\tilde{\mathcal{T}}_\pi^k v_\pi - v_\pi\|$$
$$\leq \gamma^k \|\tilde{\mathcal{T}}_\pi^{(n-1)k} v_\pi - v_\pi\| + \epsilon.$$

By repeatedly applying this inequality we can obtain:

$$\|\tilde{\mathcal{T}}_\pi^{nk} v_\pi - v_\pi\| \leq \sum_{t=0}^{n-1} \epsilon \cdot \gamma^{(tk)} = \epsilon \cdot \frac{1 - \gamma^{nk}}{1 - \gamma^k}.$$

Next, from the continuity of $\|\cdot\|$, we can take limits to obtain:

$$\epsilon \cdot \frac{1}{1-\gamma^k} \geq \lim_{n \to \infty} \|\tilde{\mathcal{T}}_\pi^{nk} v_\pi - v_\pi\| = \|\lim_{n \to \infty} \tilde{\mathcal{T}}_\pi^{nk} v_\pi - v_\pi\| = \|\tilde{v}_\pi - v_\pi\|,$$

giving us that $\mathcal{M}^k(\{\pi\}, \{v_\pi\}; \epsilon) \subseteq \mathcal{M}^\infty(\{\pi\}; \epsilon \cdot \frac{1}{1-\gamma^k})$. We can plug this result back into Eq. 23 to obtain:

$$\mathcal{M}^k(\Pi, \mathcal{V}; \epsilon) \subseteq \bigcap_{\pi \in \Pi} \mathcal{M}^k(\{\pi\}, \{v_\pi\}; \epsilon) \subseteq \bigcap_{\pi \in \Pi} \mathcal{M}^\infty(\{\pi\}; \epsilon \cdot \frac{1}{1-\gamma^k}) = \mathcal{M}^\infty(\Pi; \epsilon \cdot \frac{1}{1-\gamma^k}),$$

as needed. $\qquad\square$

**Proposition 4.** *For any set of policies $\Pi \subseteq \mathbb{\Pi}$, set of functions $\mathcal{V} \in \mathbb{V}$, $c > 1$ and error $\epsilon \in \bar{\mathbb{R}}^+$, we have*

$$\mathcal{M}^k(\Pi, c\text{-vspan}(\mathcal{V}); \epsilon) \subseteq \mathcal{M}^k(\Pi, \mathcal{V}; \epsilon) \subseteq \mathcal{M}^k(\Pi, c\text{-vspan}(\mathcal{V}); c \cdot \epsilon). \tag{13}$$

*Proof.* Clearly, $\mathcal{V} \subseteq c\text{-vspan}(\mathcal{V})$ and thus $\mathcal{M}^k(\Pi, c\text{-vspan}(\mathcal{V}); \epsilon) \subseteq \mathcal{M}^k(\Pi, \mathcal{V}; \epsilon)$. We now prove that $\mathcal{M}^k(\Pi, \mathcal{V}; \epsilon) \subseteq \mathcal{M}^k(\Pi, \mathcal{V}; c \cdot \epsilon)$. We first consider any $\tilde{m} \in \mathcal{M}^k(\Pi, \mathcal{V}; \epsilon)$ and $v' \in c\text{-vspan}(\mathcal{V})$. Since $v' \in c\text{-vspan}(\mathcal{V})$ we can write $v' = \sum_{i=1}^n \alpha_i v_i$ where $v_i \in \mathcal{V}$ for each $i$ and $\sum_{i=1}^n |\alpha_i| \leq c$. From here we observe:

$$\begin{aligned}
\|\tilde{\mathcal{T}}_\pi^k v' - \mathcal{T}_\pi^k v'\| &= \|\tilde{\mathcal{T}}_\pi^k (\sum_{i=1}^n \alpha_i v_i) - \mathcal{T}_\pi^k (\sum_{i=1}^n \alpha_i v_i)\| \\
&\leq \|\sum_{i=1}^n \alpha_i (\tilde{\mathcal{T}}_\pi^k v_i - \mathcal{T}_\pi^k v_i)\| \\
&\leq \sum_{i=1}^n |\alpha_i| \|\tilde{\mathcal{T}}_\pi^k v_i - \mathcal{T}_\pi^k v_i\| \\
&\leq \sum_{i=1}^n |\alpha_i| \epsilon \\
&\leq c \cdot \epsilon
\end{aligned} \tag{24}$$

which shows that $\mathcal{M}^k(\Pi, c\text{-vspan}(\mathcal{V}); c \cdot \epsilon)$ as needed. $\qquad\square$

**Corollary 2.** *When either $c = 1$ or $\epsilon = 0$, for any $\Pi \subseteq \mathbb{\Pi}$, $\mathcal{V} \subseteq \mathbb{V}$ it follows that*

$$\mathcal{M}^k(\Pi, \mathcal{V}; \epsilon) = \mathcal{M}^k(\Pi, c\text{-vspan}(\mathcal{V}); \epsilon). \tag{14}$$

*Proof.* The proof follows directly from Proposition 4. When either $c \in \{0, 1\}$ the left-most and right-most terms in Eq. 13 are equal, squeezing $\mathcal{M}^k(\Pi, \mathcal{V}; \epsilon) = \mathcal{M}^k(\Pi, c\text{-vspan}(\mathcal{V}); \epsilon)$ as needed. $\qquad\square$

**Proposition 5.**

1. **(Asymmetry)** *For any $\mathcal{V} \subseteq \mathcal{V}' \subseteq \mathcal{V}'' \subseteq \mathbb{V}$ it follows that*

$$0 = \beta(\mathcal{V}||\mathcal{V}) \leq \beta(\mathcal{V}||\mathcal{V}') \leq \beta(\mathcal{V}||\mathcal{V}'') \quad and \quad 0 = \beta(\mathcal{V}''||\mathcal{V}'') \leq \beta(\mathcal{V}'||\mathcal{V}'') \leq \beta(\mathcal{V}||\mathcal{V}'').$$

2. **(Convex, Compact $\mathcal{V}$)** *When $\mathcal{V}$ is convex and compact it follows that*

$$\beta(\mathcal{V}||\mathcal{V}') = \beta(\mathcal{V}||1\text{-vspan}(\mathcal{V}')).$$

*Proof.*

1. Recall $\beta(\mathcal{V}||\mathcal{V}') = \max_{v' \in \mathcal{V}'} \min_{v \in \mathcal{V}} \|v' - v\|$. Increasing the size of $\mathcal{V}'$ means that more elements can be maximized over, thereby increasing $\beta(\mathcal{V}||\mathcal{V}')$. Similarly, increasing the size of $\mathcal{V}$ means that more elements can be minimized over, thereby decreasing $\beta(\mathcal{V}||\mathcal{V}')$. When $\mathcal{V} = \mathcal{V}'$, we know that

$$0 \leq \beta(\mathcal{V}||\mathcal{V}') = \max_{v' \in \mathcal{V}'} \min_{v \in \mathcal{V}} \|v' - v\| \leq \max_{v' \in \mathcal{V}'} \|v' - v'\| = 0,$$

where second inequality follows since $\mathcal{V} = \mathcal{V}'$.

2. We begin by considering the function $g(v') = \min_{v \in \mathcal{V}} \|v - v'\|$. We begin by showing that this function is convex. Consider $v_1', v_2' \in \mathcal{V}'$ and denote $v_1 = \text{argmin}_{v \in \mathcal{V}} \|v - v_1'\|$ and $v_2 = \text{argmin}_{v \in \mathcal{V}} \|v - v_2'\|$. Then for any $\lambda \in [0, 1]$ we can write:

$$
\begin{aligned}
\lambda g(v_1') + (1 - \lambda)g(v_2') &= \lambda\|v_1' - v_1\| + (1 - \lambda)\|v_2' - v_2\| \\
&\geq \|(\lambda v_1' + (1 - \lambda)v_2') - (\lambda v_1 + (1 - \lambda)v_2)\|
\end{aligned}
\tag{25}
$$

since $\mathcal{V}$ is convex $(\lambda v_1 + (1 - \lambda)v_2) \in \mathcal{V}$, thus:

$$
\begin{aligned}
\|(\lambda v_1' + (1 - \lambda)v_2') - (\lambda v_1 + (1 - \lambda)v_2)\| &\geq \min_{v \in \mathcal{V}} \|(\lambda v_1' + (1 - \lambda)v_2') - v\| \\
&= g(\lambda v_1' + (1 - \lambda)v_2')
\end{aligned}
\tag{26}
$$

which suffices to show that $g$ is a convex function.

Next we consider any element $v' \in 1\text{-vspan}(\mathcal{V}')$ such that $v' = \sum_i \alpha_i v_i'$ with $\sum_i \alpha_i = 1$ and $\alpha_i \geq 0$ for all $i$. We can then write:

$$g(v') = g(\sum_i \alpha_i v_i') \leq \sum_i \alpha_i g(v_i') \leq \max_i g(v_i') \leq \max_{v' \in \mathcal{V}'} g(v') = \beta(\mathcal{V}||\mathcal{V}') \tag{27}$$

Since $g(v') \leq \beta(\mathcal{V}||\mathcal{V}')$ for every $v' \in 1\text{-vspan}(V')$ it then follows that

$$\beta(\mathcal{V}||1\text{-vspan}(\mathcal{V}')) = \max_{v' \in 1\text{-vspan}(\mathcal{V}')} g(v') \leq \beta(\mathcal{V}||\mathcal{V}'). \tag{28}$$

We obtain the reverse equality by noting that $\mathcal{V}' \subseteq 1\text{-vspan}(\mathcal{V}')$ and thus $\beta(\mathcal{V}||\mathcal{V}') \leq \beta(\mathcal{V}||1\text{-vspan}(\mathcal{V}'))$. Hence $\beta(\mathcal{V}||1\text{-vspan}(\mathcal{V}')) = \beta(\mathcal{V}||\mathcal{V}')$ as needed.

$\square$

**Proposition 6.** *For any $\Pi \in \overline{\Pi}$, $\mathcal{V}, \mathcal{V}' \in \mathbb{V}$ and $\epsilon \in \bar{\mathbb{R}}^+$, it follows that*

$$\mathcal{M}^k(\Pi, \mathcal{V}; \epsilon) \subseteq \mathcal{M}^k(\Pi, \mathcal{V}'; \epsilon + 2\gamma^k \beta(\mathcal{V}||\mathcal{V}')),$$

*moreover, if $\mathcal{V}$ is convex and compact, we obtain:*

$$\mathcal{M}^k(\Pi, \mathcal{V}; \epsilon) \subseteq \mathcal{M}^k(\Pi, 1\text{-vspan}(\mathcal{V}'); \epsilon + 2\gamma^k \beta(\mathcal{V}||\mathcal{V}')).$$

*Proof.* Fix an arbitrary model $\tilde{m} \in \mathcal{M}^k(\Pi, \mathcal{V}; \epsilon)$ and any $\pi \in \Pi$. We now select some $v' \in \mathcal{V}'$ and examine the tolerance with which $\tilde{m}$ is value equivalent with respect to $\{\pi\}$ and $\{v'\}$.

Notice that for *any* $v \in \mathcal{V}$ we can write

$$
\begin{aligned}
\|\tilde{\mathcal{T}}_\pi^k v' - \mathcal{T}_\pi^k v'\| &= \|\tilde{\mathcal{T}}_\pi^k v' - \mathcal{T}_\pi^k v' + \tilde{\mathcal{T}}_\pi^k v - \tilde{\mathcal{T}}_\pi^k v\| \\
&\leq \|\tilde{\mathcal{T}}_\pi^k v' - \tilde{\mathcal{T}}_\pi^k v\| + \|\tilde{\mathcal{T}}_\pi^k v - \mathcal{T}_\pi^k v'\| \\
&= \|\tilde{\mathcal{T}}_\pi^k v' - \tilde{\mathcal{T}}_\pi^k v\| + \|\tilde{\mathcal{T}}_\pi^k v - \mathcal{T}_\pi^k v' + \mathcal{T}_\pi^k v - \mathcal{T}_\pi^k v\| \\
&\leq \|\tilde{\mathcal{T}}_\pi^k v' - \tilde{\mathcal{T}}_\pi^k v\| + \|\tilde{\mathcal{T}}_\pi^k v - \mathcal{T}_\pi^k v\| + \|\mathcal{T}_\pi^k v - \mathcal{T}_\pi^k v'\| \\
&\overset{(1)}{\leq} 2\gamma^k \|v' - v\| + \|\tilde{\mathcal{T}}_\pi^k v - \mathcal{T}_\pi^k v\| \\
&\overset{(2)}{\leq} 2\gamma^k \|v' - v\| + \epsilon
\end{aligned}
\tag{29}
$$

where (1) follows from the Bellman operators $\tilde{\mathcal{T}}_\pi$ and $\mathcal{T}_\pi$ being contractions and (2) follows the assumption that $\tilde{m} \in \mathcal{M}^k(\Pi, \mathcal{V})$.

Since the above upper bound on $\|\tilde{\mathcal{T}}_\pi^k v' - \mathcal{T}_\pi^k v'\|$ holds for any $v \in \mathcal{V}$ we can write that

$$
\|\tilde{\mathcal{T}}_\pi^k v' - \mathcal{T}_\pi^k v'\| \leq \epsilon + 2\gamma^k \min_{v \in \mathcal{V}} \|v' - v\|.
\tag{30}
$$

Thus far we have shown that $\mathcal{M}^k(\Pi, \mathcal{V}; \epsilon) \subseteq \mathcal{M}^k(\Pi, \{v'\}; \epsilon + 2\gamma^k \min_{v \in \mathcal{V}} \|v' - v\|)$. To find a tolerance that holds for all $v' \in \mathcal{V}'$ we simply take a maximum over the element-wise tolerance:

$$
\max_{v' \in \mathcal{V}} \epsilon + 2\gamma^k \min_{v \in \mathcal{V}} \|v' - v\| = \epsilon + 2\gamma^k \beta(\mathcal{V} \| \mathcal{V}')
\tag{31}
$$

This completes the proof. $\qquad \square$

**Theorem 2.** *For any $\tilde{m} \in \mathcal{M}^k(\mathbb{\Pi}, \mathcal{V}; \epsilon)$ it follows that*

$$
\|v_* - v_{\tilde{\pi}_*}\| \leq \frac{2}{1-\gamma^k} \cdot \min_{c \geq 1} \left( c \cdot \epsilon + 2\gamma^k \beta(c\text{-vspan}(\mathcal{V}) \| \mathbb{V}_{\mathbb{\Pi}}) \right),
\tag{17}
$$

*where $\tilde{\pi}_*$ is an optimal policy of $\tilde{m}$.*

*Proof.* From Theorem 1, we know by tolerating an error of

$$
\epsilon' = \frac{1}{1-\gamma^k} \min_{c \geq 1} (c \cdot \epsilon + 2\gamma^k \beta(c\text{-vspan}\mathcal{V}) \| \mathbb{V}_{\mathbb{\Pi}})),
$$

that $\mathcal{M}^k(\mathbb{\Pi}, \mathcal{V}; \epsilon) \subseteq \mathcal{M}^\infty(\mathbb{\Pi}; \epsilon')$. Thus $\mathcal{E}_\epsilon(\mathbb{\Pi}, \mathcal{V}, k \,|\, \mathbb{\Pi}, \infty) \leq \epsilon'$. By applying Proposition 2, we obtain $\|v_* - v_{\tilde{\pi}_*}\| \leq 2\epsilon'$ as needed. $\qquad \square$

**Corollary 3.** *Let $\hat{\mathbb{V}}_{\mathbb{\Pi}} = \{\hat{v}_\pi : \pi \in \mathbb{\Pi}\}$ be a set of approximate value functions satisfying $\|v_\pi - \hat{v}_\pi\| \leq \epsilon_{approx}$ for all $\pi \in \mathbb{\Pi}$. Then for any $\tilde{m} \in \mathcal{M}^k(\mathbb{\Pi}, \hat{\mathbb{V}}_{\mathbb{\Pi}}; \epsilon)$ it follows that:*

$$
\|v_* - v_{\tilde{\pi}_*}\| \leq \frac{2(\epsilon + 2\gamma^k \epsilon_{approx})}{1 - \gamma^k},
$$

*where $\tilde{\pi}_*$ is any optimal policy in $\tilde{m}$.*

*Proof.* From the definition of $\hat{\mathbb{V}}_{\mathbb{\Pi}}$, we know that $\beta(\mathbb{V}_{\mathbb{\Pi}} \| \hat{\mathbb{V}}_{\mathbb{\Pi}}) \leq \epsilon_{approx}$. Thus by Proposition 6 and Corollary 1 we know that

$$
\mathcal{M}^k(\mathbb{\Pi}, \hat{\mathbb{V}}_{\mathbb{\Pi}}; \epsilon) \subseteq \mathcal{M}^k(\mathbb{\Pi}, \mathbb{V}_{\mathbb{\Pi}}; \epsilon + 2\gamma^k \epsilon_{approx}) \subseteq \mathcal{M}^\infty(\mathbb{\Pi}; \frac{\epsilon + 2\gamma^k \epsilon_{approx}}{1 - \gamma^k}),
$$

thus $\mathcal{E}_\epsilon(\mathbb{\Pi}, \hat{\mathbb{V}}_{\mathbb{\Pi}}, k \,|\, \mathbb{\Pi}, \infty) \leq \frac{\epsilon + 2\gamma^k \epsilon_{approx}}{1 - \gamma^k}$, which gives us the desired performance bound by an application of Theorem 2. $\qquad \square$