# OpenReview forum: "Approximate Value Equivalence"
_NeurIPS.cc/2022/Conference — NeurIPS 2022 Accept_

### Official Review · Reviewer_3UJe · 2022-07-11

**Rating:** 4
**Confidence:** 3
**Soundness:** 4 excellent
**Presentation:** 3 good
**Contribution:** 2 fair

**Summary:**

This paper extends the notion of "value equivalence" among MDPs w.r.t. sets of policies and value functions (and number of steps) to "approximate value equivalence," that permits the values after k steps of the policy to differ by  up to some parameter epsilon. It is then possible, for any two classes of models, to find an epsilon for which the models satisfy the definition. Bounds that relate the performance of policies across such approximate value equivalent models are also obtained.


**Questions:**

Can you point to evidence that it is productive to consider this notion? (See the last part of "Strengths and Weaknesses" for a couple examples of what this might look like.)

**Limitations:**

This is fine.

**Strengths And Weaknesses:**

It seems clear that for the kind of models studied here, we need bounds that control how close the predicted performance is to performance in the actual environment, and it seems desirable to be able to obtain this for bounded orders. This is the main strength of the paper.

The notion of approximate value equivalence is a very natural generalization of the notion of value equivalence -- indeed, it borders on being "obvious" given the existing notion of value equivalence. I am not so moved by the results about relating arbitrary pairs of classes. I don't see the utility of this notion for such large values of epsilon as such relationships would require -- and indeed, intuitively, we would not expect totally arbitrary pairs to be "equivalent" in any meaningful sense. This dampens the significance of these contributions.

What's really missing from this work is evidence that the definition can be made to do useful work for us. I'd want to see it used to analyze an algorithm or show that some MDP can be represented more compactly, or something of similar flavor.

---

> ### Author Response · Authors · 2022-08-02
> **Response to 3UJe**
>
> We would like to thank the reviewer for their thoughtful comments on our submission.
>
> **It seems clear that for the kind of models studied here, we need bounds that control how close the predicted performance is to performance in the actual environment, and it seems desirable to be able to obtain this for bounded orders. This is the main strength of the paper.**
>
> As the reviewer points out, it is important to be able to quantify how well a learned value equivalent (VE) model will perform when used for planning. To our knowledge, ours is the first work to address this problem. Previous work did not present any performance bounds, but rather (restrictive) conditions under which VE models would perform optimally.
>
> In particular, [1] showed that a VE model with respect to all policies and all functions must also be the true model of the environment, and thus yields optimal planning. This result was generalized in [2], in which it was shown that any proper VE model with respect to all deterministic policies also yields optimal planning.
>
> Besides these two quite restrictive scenarios, there are no results on the performance of VE models available in the literature. To illustrate how restrictive this is, note that if a single function or a single policy is removed from either of the aforementioned model classes, we have no guarantees whatsoever on the performance of the models within the resulting class.
>
> This is the gap in the literature that our work aims to bridge: to provide bounds on the planning performance of VE model classes with arbitrary orders and function sets.
>
> In hindsight, we feel like we should have made this point more clear in the paper, and we intend to do so in the revised version.
>
> **I am not so moved by the results about relating arbitrary pairs of classes. I don't see the utility of this notion for such large values of epsilon as such relationships would require -- and indeed, intuitively, we would not expect totally arbitrary pairs to be "equivalent" in any meaningful sense.**
>
> The reviewer is correct in that the bounds provided in our submission can be quite loose when general VE model classes are compared. Note though that the ability to compare general VE classes is not an end in itself, but rather a means to convert the topological properties presented in previous work into performance bounds, which were nonexistent up until now.
>
> **What's really missing from this work is evidence that the definition can be made to do useful work for us. I'd want to see it used to analyze an algorithm or show that some MDP can be represented more compactly, or something of similar flavor. Can you point to evidence that it is productive to consider this notion?**
>
> We agree that our submission should have been more clear regarding the benefits of resorting to the notion of approximate VE (AVE). We have updated the paper to fix that. Here, we summarize why we think it is productive to think in terms of AVE:
>
> 1) As pointed out above, the concept of AVE allows us to provide performance guarantees for VE classes of any order and with respect to any set of functions. This sort of guarantee was not presented in previous work, which focused on idealized scenarios that are useful to present a new concept but completely disconnected from the practice. This point should have been emphasized more in the paper, and we will make sure it is clear in the revised version (as can be seen in the version currently submitted).
>
> 2) Past work [1,2] have assumed, either implicitly or explicitly, that increasing the number of functions used to define a VE class is always beneficial. However, this was not always corroborated by empirical evidence, since in some experiments increasing the number of functions would in fact harm performance. Using the AVE formalism, we were able to present a more nuanced view that clearly characterizes the trade-offs involved in adding more functions and describes the empirical evidence more accurately. Concretely, AVE makes it clear why sometimes it is preferable to be VE with respect to a smaller set of functions: because this will in general yield a small VE error (epsilon). We provide instantiations of this phenomenon both in the form of examples and small experiments in the newly added Section 6 of the updated submission.
>
> [1] Grimm, Christopher, et al. "The value equivalence principle for model-based reinforcement learning." Advances in Neural Information Processing Systems 33 (2020): 5541-5552.
>
> [2] Grimm, Christopher, et al. "Proper value equivalence." Advances in Neural Information Processing Systems 34 (2021): 7773-7786.

---

### Official Review · Reviewer_pwPJ · 2022-07-11

**Rating:** 7
**Confidence:** 4
**Soundness:** 3 good
**Presentation:** 3 good
**Contribution:** 3 good

**Summary:**

This paper proposes approximate value equivalence (AVE) which extends the previously proposed VE formalism by replacing equialities with error tolerances. With this extension, it shows that AVE models with respect to one set of functions are also AVE with respect to any other set of functions if a high enough error is tolerated, which then allows for deriving bounds on the performance of VE models with respect to arbitrary sets of functions by relating them to a particular model class with known performance guarantees. Finally, the paper supports these results by providing intuitions and discussions of their implications.

**Questions:**

**Major:**

1. On the role of the deterministic policy set in PVE classes: Why does the PVE class wrt all deterministic policies contains only models which can plan optimally (see the last par. of Sec. 2)? I mean what is the role of determinism here? Why doesn’t the PVE class wrt all policies (including deterministic and stochastic ones) contains only models which can plan optimally?
2. On the significance of the theoretical results in terms of practicality: In the previous studies of Grimm et al. (on VE), the authors have provided experimental results that either justify the usage of VE models or improve the currently existing algorithms. However, in this study there is no empirical demonstration of the usefulness of the theoretical results. Even though the theoretical results are interesting by themselves, I think that providing a section on how these results can at least help developing better VE MBRL algorithms can significantly benefit the paper. Is it possible to provide a discussion on how these theoretical results can be useful in giving rise to new algorithms?

**Minor:**

1. In Def. 1 and Property 1, $\mathcal{V}$ should be a subset of $\mathbb{V}$?
2. I think the par. that starts with line 164 should explicitly mention that a certain amount of error will be tolerated for $\mathcal{M}^{\infty}$. I had to read it several times to get this part.
3. In Prop. 2 should it be $\mathbb{V}_{\mathbb{\Pi}}$ in the upper bound?
4. Why is there $2^{\mathbb{V}}$’s in line 242? Shouldn’t these be $\mathbb{V}$’s?
5. There is a typo at line 257 right after the second column.
6. There seems to be a typo inside beta in Eq. 18 where one of the value functions should not have a hat?

If the authors are able to address the concerns provided in this review during the rebuttal period, I am willing to raise my score.


**Limitations:**

Yes the authors mention the limitations of their work.

**Strengths And Weaknesses:**

**Originality:** The paper seems to be original in the sense that it extends the previously proposed VE formalism by replacing equialities with error tolerances and that it develops a whole variety of new theoretical results out of this. I am not aware of any study that does this. The related work section also seems to cover the related studies in the literature.

**Quality:** Overall, there seems to be no serious quality issues with the paper. Though, I would like to indicate that I haven’t checked the proofs in a very detailed manner and just went over them at a high-level. The submission looks technically sound and the core ideas are well-explained. However, I do have a few major and (mostly) minor questions on the quality of the paper (see the Questions section below).

**Clarity:** I found no problems with the clarity of the paper. The paper is very well-written. I have nothing to suggest.

**Significance:** Even though the derived theoretical results seem to be interesting in their own sense, I would like to indicate that I found the paper to have problems in the significance of the results. More specifically, I am not sure if the paper provides enough discussion on how the derived results can be helpful in developing VE MBRL algorithms. More on this can be found in Question 2 below.

---

> ### Author Response · Authors · 2022-08-02
> **Response to pwPJ**
>
> We would like to thank you for taking the time to review our paper (we are also grateful for your attention to detail which helps us catch and correct typos!)
>
> **On the role of the deterministic policy set in PVE classes: Why does the PVE class wrt all deterministic policies contains only models which can plan optimally (see the last par. of Sec. 2)? I mean what is the role of determinism here? Why doesn’t the PVE class wrt all policies (including deterministic and stochastic ones) contains only models which can plan optimally?**
>
> This is a good question!
>
> Both the PVE class with respect to all policies and the PVE class with respect to all deterministic policies contain only models which plan optimally, but the former class is a subset of the latter. Accordingly, [2] showed that the PVE class with respect to all deterministic policies contains only models which plan optimally (see Corollary 1 in [2]); note that the result also holds for the PVE class with respect to all policies (which is a subset of the previous one). Our reason for mentioning the PVE model class wrt all deterministic policies was to point out Grimm et al’s most general result about model classes that only contain models which can plan optimally.
>
> **On the significance of the theoretical results in terms of practicality: In the previous studies of Grimm et al. (on VE), the authors have provided experimental results that either justify the usage of VE models or improve the currently existing algorithms. However, in this study there is no empirical demonstration of the usefulness of the theoretical results. Even though the theoretical results are interesting by themselves, I think that providing a section on how these results can at least help developing better VE MBRL algorithms can significantly benefit the paper. Is it possible to provide a discussion on how these theoretical results can be useful in giving rise to new algorithms?**
>
> We agree that the original submission was missing a demonstration of the utility of the theoretical results. Accordingly, we have updated the submission to include a new section (Section 6) which shows how approximate value equivalence can facilitate in the design of VE MBRL algorithms.
>
> In particular, we consider a setting where an agent with a limited capacity model can choose between learning a model with respect to many functions (and tolerate a high approximation error due to having low-capacity) or learning a model with respect to fewer functions (and tolerate a lower approximation error). We showed in our original submission that higher approximation error results in worse performance guarantees (Proposition 2), so it is natural to wonder if, in such settings, it is ever preferable for the agent to choose to learn a VE model with respect to fewer functions.  We empirically show that there are indeed situations like this, hinting at the utility of AVE in deciding what types of value equivalent models to learn.
>
> **In Def. 1 and Property 1,  should be a V subset of \VS?**
>
>
> **There seems to be a typo inside beta in Eq. 18 where one of the value functions should not have a hat?**
>
> **In Prop. 2 should it be  \FS_\PS in the upper bound?**
>
>
> Good finds! We have fixed all of the above in the updated submission.
>
> I think the par. that starts with line 164 should explicitly mention that a certain amount of error will be tolerated for M^inf. I had to read it several times to get this part.
> This is a good point, we’ve made this explicit in the revised submission.
>
> **Why is there 2^V’s in line 242? Shouldn’t these be V’s?**
>
> We believe this is correct. Beta takes as input 2 arguments each of which may be any subset of FS. This means that each argument is a member of the power-set of FS denoted 2^FS. We have added a note in the submission about this notation to improve clarity.
>
> [1] Grimm, Christopher, et al. "The value equivalence principle for model-based reinforcement learning." Advances in Neural Information Processing Systems 33 (2020): 5541-5552.
>
> [2] Grimm, Christopher, et al. "Proper value equivalence." Advances in Neural Information Processing Systems 34 (2021): 7773-7786.

---

> > ### Comment · Reviewer_pwPJ · 2022-08-08
> > **Response to the Authors**
> >
> > I would like to thank the authors for providing detailed answers to the questions that I had. I have no further question and I believe that with the addition of Section 6, the paper is in a better shape. I have updated my score accordingly.

---

### Official Review · Reviewer_bf62 · 2022-07-12

**Rating:** 6
**Confidence:** 2
**Soundness:** 3 good
**Presentation:** 4 excellent
**Contribution:** 2 fair

**Summary:**

The paper discusses the notion of _approximate value equivalence_ in model-based reinforcement learning. Approximate value equivalence is concerned with the study of families of models that yield $k$-step Bellman updates that are approximately equivalent to the Bellman update yielded by the true model, in the sense that---given a policy $\pi$ in some set of policies and a value function $v$ in some set of functions---$\tilde{\mathcal{T}}^k_\pi v\approx\mathcal{T}^k_\pi v$ (here $\tilde{\mathcal{T}}_\pi$ and $\mathcal{T}_\pi$ represent, respectively, the approximate and exact Bellman operators). The paper provides a number of results that relate different families of models in terms of the error that they incur in terms of Bellman updates.

**Questions:**

My main question is related with the practical use of the bounds provided in the paper, in light of the difficulty of determining the factor dependent on the distance/divergence $\beta$, and the minimum tolerated error $\mathcal{E}_\epsilon$.

**Limitations:**

N/A.

**Strengths And Weaknesses:**

*Strong points:*
* The problem addressed (understanding how approximate models impact the error incurred in the computation of the value function in RL) is important and interesting;
* The presentation is very clear, well-organized, and several of the results are quite intuitive.

*Weak points:*
* The article is not self-contained, in the sense that most of the less intuitive results are presented with no hint on the proof. The paper still has available 1/2 page that could be used to provide even if a one-liner hint on how the result is established.
* It is not clear what's the practical use of the proposed bounds. The authors make an effort in Section 5 to discuss this issue, but I believe that this is a core point in assessing the significance of the contributions of the paper. In particular, the provided bounds depend critically on the distance $\beta$ and the minimum tolerated error $\mathcal{E}_\epsilon$, but---as far as I can understand---these are very difficult to assess in practical scenarios (particular in scenarios involving high-capacity approximations). Therefore, it would be important to discuss, in light of this, the practical use of the provided bounds.

---

> ### Author Response · Authors · 2022-08-02
> **Response to bf62**
>
> Thank you for your careful review, we appreciate the thoughtful comments!
>
> **The article is not self-contained, in the sense that most of the less intuitive results are presented with no hint on the proof. The paper still has available 1/2 page that could be used to provide even if a one-liner hint on how the result is established.**
>
> We agree that more intuition could have been provided in some of the results, and will amend that in the revised version of the paper. If the reviewer has specific suggestions about where intuition would be mostly welcome please let us know and we will be happy to add it to the paper.
>
> **It is not clear what's the practical use of the proposed bounds. The authors make an effort in Section 5 to discuss this issue, but I believe that this is a core point in assessing the significance of the contributions of the paper.**
>
> We agree that a discussion of the practical use of the bounds is missing from the original submission and have provided an additional section (Section 6) discussing this, including an experiment.
>
> In particular, we show that when model capacity is limited, there are situations where an agent can choose between learning a VE model with respect to more functions (and tolerate a higher error) or learning a VE model with respect to fewer functions (and tolerate a lower error). In our submission we showed that tolerating a higher error results in poorer performance guarantees (Theorem 2) for the associated VE models, so it is natural to wonder if there are ever situations where it is better to learn a VE model with respect to fewer functions (note that this is a nuance that has gone unexplored in the previous works on VE [1,2], which encouraged learning VE / PVE models with respect to as many functions as was practical).
>
> **In particular, the provided bounds depend critically on the distance  and the minimum tolerated error , but---as far as I can understand---these are very difficult to assess in practical scenarios (particular in scenarios involving high-capacity approximations).**
>
> **My main question is related with the practical use of the bounds provided in the paper, in light of the difficulty of determining the factor dependent on the distance/divergence , and the minimum tolerated error .**
>
> The reviewer is correct to point out that the minimum tolerated error defined in Definition 2 is very difficult to assess. However, we don’t need to compute the minimum tolerated error, just an upper-bound on it. In Proposition 2, we show that we can bound the suboptimality of a model’s planning performance in terms of its minimum tolerated error (or any upper bound on it).
>
> This is one of the core contributions of our paper: a framework for converting properties about the topology of AVE classes (minimum tolerated errors or upper-bounds thereof) into results about the performance of models contained inside them.
>
> In the subsequent sections of the paper we provide several upper-bounds on this error (Propositions 3, 4 and 6) which apply when computing the minimum tolerated error between different types of VE classes. We would like to emphasize that our framework facilitates the future study of these bounds.
>
> [1] Grimm, Christopher, et al. "The value equivalence principle for model-based reinforcement learning." Advances in Neural Information Processing Systems 33 (2020): 5541-5552.
>
> [2] Grimm, Christopher, et al. "Proper value equivalence." Advances in Neural Information Processing Systems 34 (2021): 7773-7786.

---

> > ### Comment · Reviewer_bf62 · 2022-08-10
> > **Thank you for your response**
> >
> > I thank the authors for the response to the issues I raised. Some additional notes:
> >
> > * Regarding providing some intuition on how the different propositions are established, I think that a 1-liner hint on how the result is established would be helpful in all, but - personally - I found that Propositions 2 and 3 could use that 1-liner intuition about the proof.
> >
> > * I appreciate the inclusion of Section 6, which I believe sheds some light regarding the potential practical significance the results in the paper.
> >
> > * Regarding the usefulness of the bounds provided in the paper, I acknowledge the author's reply. I am still not fully convinced regarding how easily meaningful bounds can be derived from the topological properties of the AVE classes, but I agree that the results in the paper provide an important link between these, given $\beta$ and $\mathcal{E}_\epsilon$.

---

### Author Response · Authors · 2022-08-02
**General Response to Reviewers**

We sincerely thank all the reviewers for their careful reviews and constructive feedback! We will address each review individually, but here we make some general comments that apply to all the reviews, and present a summary of the changes made to the paper.

First, we want to address a comment that has shown up in all of the reviews: the practical “use” of the notion of approximate value equivalence (AVE). In short, the AVE formalism allows us to provide performance guarantees for broad families of VE classes. Such guarantees were not available in the literature, which focused on idealized scenarios that do not reflect the practice. In particular, in previous work guarantees were only provided for two very restrictive cases: when all the policies and functions are used to enforce VE [1] or when all deterministic policies are used to enforce proper VE [2]. Our paper extends these guarantees to VE classes of any order and with respect to any set of functions.

Still on the practical benefits of AVE, we notice that previous work [1,2] painted a somewhat simplistic picture of VE in which more functions or policies were always beneficial. In this paper we present a more nuanced view that highlights the fact that, when the model has limited capacity, sometimes fewer functions yields better performance. We added a new section to the paper that discusses this point in more depth (Section 6). The new section includes a set of experiments that nicely illustrate the trade-offs involved in using AVE in practice, including the fact that more functions does not always yield improved performance.

[1] Grimm, Christopher, et al. "The value equivalence principle for model-based reinforcement learning." Advances in Neural Information Processing Systems 33 (2020): 5541-5552.

[2] Grimm, Christopher, et al. "Proper value equivalence." Advances in Neural Information Processing Systems 34 (2021): 7773-7786.

---

### Author Response · Authors · 2022-08-07
**We value your feedback**

We would like to thank the reviewers again for their feedback. We are committed to producing the best possible version of our submission, and have already included a new section with experiments to address a shared concern among the reviewers regarding the practical applicability of our theoretical results. We are willing to interactively address any remaining concerns, so please let us know if there is anything else we can do to improve our submission.

---

### Meta-Review · Area_Chair_JpGJ · 2022-08-28

**Recommendation:** Accept
**Confidence:** Less certain

**Metareview:**

A key discussion point in the rebuttal phase was the practical use of the proposed bounds, which two of the three reviewers brought up. The authors in response added an additional section (Section 6) and experiment to address this concern. While some concerns regarding the practical use of these bounds remain, the authors have made a sufficiently convincing case in my view. Hence, I recommend acceptance.

**Award:**

No

---

### Decision · Program_Chairs · 2022-09-14

Accept